# Evidence into practice: protocol for a new mixed-methods approach to explore the relationship between trials evidence and clinical practice through systematic identification and analysis of articles citing randomised controlled trials

Benjamin E Byrne,[1] Leila Rooshenas,[1] Helen Lambert,[2] Jane M Blazeby[1]

¹Centre for Surgical Research, Population Health Sciences, Bristol Medical School, University of Bristol, Bristol, UK
²Population Health Sciences, Bristol Medical School, University of Bristol, Bristol, UK

**Correspondence to**
Benjamin E Byrne;
benbyrne@doctors.org.uk

## ABSTRACT

**Introduction** Randomised controlled trials (RCTs) provide high-quality evidence to inform practice. However, much routine care is not based on available RCT evidence. Understanding this disconnect may improve trial design, reporting and implementation. Published literature commenting on RCTs may yield relevant insights. This protocol presents a new approach examining how researchers understand, contextualise and use evidence from RCTs, through analysis of letters, editorials and discussion pieces citing individual RCTs. Surgical case studies will illustrate its ability to identify wide-ranging factors influencing application of trials evidence.

**Methods and analysis** In-depth study of published literature will explore written responses to RCTs. After purposefully selecting individual RCTs, we will systematically identify all citing articles covered in Web of Science and Scopus. Editorials, discussions and letters will be included. These are considered most likely to provide critiques and opinions about index RCTs. Original articles and reviews will be excluded. Clinical specialty, RCT design, outcomes and bibliographical data will be collected for RCTs and citing articles. Citing articles will be thematically analysed using the constant comparison technique to explore author understanding, contextualisation and relationship to clinical practice for the index trial. Coding will include generic issues relevant to all RCTs, such as sample size or blinding, and features specific to surgery, such as learning curve. Index trial quality will be examined using validated tools. Results will be combined to create a broad overview of the understanding and use of RCT evidence.

**Ethics and dissemination** This study involves secondary use of existing articles and does not require ethical approval. Pilot work will establish its feasibility and inform progression to larger scale utilisation across a broad range of RCTs. Findings will be published in a peer-reviewed journal and presented at surgical and methodological conferences. Results will guide future work on trial design to optimise implementation of results.

## Strengths and limitations of this study

► This new method will use published response to randomised controlled trials (RCTs) to identify how clinicians and academics interpret and use evidence from randomised trials.
► The emerging data will identify areas that need attention to improve translation of RCT results into practice.
► Rigorous qualitative analysis combined with structured data extraction on the quality and reporting of included RCTs will be used to gain in-depth understanding of the relevant issues.
► This method may have broad applicability across all randomised trials.
► The study findings may be limited by the focus on included and published articles.

## INTRODUCTION

Since the early description of evidence-based medicine,[1] it has been recognised that high-quality research is needed to improve clinical practice and patient outcomes. This typically includes well-designed and conducted randomised controlled trials (RCTs) and systematic reviews summarising the results of multiple trials in one field.[2] Such research is used to inform national guidelines which aim to influence and set standards for clinical practice.

Research across a number of areas, including surgery, has shown that day-to-day care is often not delivered in accordance with national guidelines and best evidence.[3–6] There are many reasons for lack of translation of knowledge from RCTs into practice. Trials may be compromised by problems with their design and conduct, limiting the credibility of the associated results and/

or the applicability of the trial in other settings. Internal validity can be compromised by well-known sources of bias that may be assessed with validated tools, such as the risk of bias tool.[7] Trial design and reporting may also compromise external validity, limiting the application of trial findings more broadly.[8 9] In addition, complex interventions, such as surgery, may be influenced by a wide range of factors, such as surgeon or institution caseload, teamwork, care pathways and organisational culture, which are generally referred to as 'context'.[10–12] The exact mechanisms by which such contextual elements could affect the results of care have not been defined, but they may include technical skills, decision-making, professional status that may influence quality of post-surgical care and communication. These may have clear relevance to both the efficacy and implementation of an intervention.

Other factors beyond the control or influence of trials methodologists are also important in determining whether evidence from RCTs is implemented in practice. Evidence from an individual RCT regarding a particular intervention must be considered alongside a range of other evidence types, including RCTs concerning the same or similar interventions. Clinicians, many of whom have not been involved in clinical trials, may lack familiarity with trial design; even those involved in trial recruitment may have a poor grasp of key concepts.[13 14] This, combined with evidence of poor numeracy skills,[15] may make it difficult to critically interpret and use evidence from RCTs. The culture and attitudes of clinicians are also highly relevant; in the case of surgery, there is evidence that surgeons tend to value clinical experience and individualism over research findings[16] and tend to resist changing established patterns of practice.[17]

### Proposed novel approach

Important insights into how clinicians and academics think about RCTs may be gained by examining the perspectives and opinions they convey when writing about RCTs in the published literature. Analysis of these views could improve understanding of the factors and issues of concern to clinicians when interpreting the results of an RCT and relating them to their own practice. The results may have importance across many areas, from trial design and reporting, to surgeon education, and organisational implications for surgical practice to promote uptake of effective new treatments.

This study protocol describes a new approach with the aim of establishing feasibility. It will explore the understanding, contextualisation and use of evidence from RCTs by examining existing literature that cite selected index RCTs. Surgical trials will be selected as case studies to pilot the method, to investigate its ability to capture relevant factors and potential challenges in implementation of trial findings associated with complex interventions. The results will inform progression to a larger scale project to build on preliminary findings.

**Table 1** Method outline for identifying and analysing articles citing surgical randomised controlled trials (RCTs)

| Step | Action |
|------|--------|
| 1 | Purposefully select major surgical RCTs reported in the last 10 years. |
| 2 | Identify and systematically sample the articles citing the selected RCTs. |
| 3 | Undertake in-depth qualitative analysis and identify emerging themes. |
| 4 | Summarise validity and reporting of included RCTs. |
| 5 | Combine the results of steps 3 and 4 to develop deeper understanding of how trials are understood and the relationship with trial quality. |
| 6 | Develop and refine the methodology. |
| 7 | Create recommendations to inform future design, conduct, reporting and implementation of surgical RCTs. |

## METHODS AND ANALYSIS

### Aim

This study aims to establish the feasibility of using a new method to generate insights into how evidence from RCTs is understood and assessed for potential application to clinical practice.

### Objectives

This feasibility study has several objectives: to understand how evidence from surgical RCTs is used, understood and interpreted in published articles; to assess the methodological quality of the selected RCTs; to consider the quality of the trials alongside writings about the trials to generate insights into how evidence from RCTs may, or may not, be appropriately understood and used.

### Design

The sequential steps involved in this study are summarised in table 1 and are described in the following sections.

### Purposefully select major surgical RCTs reported in the last 10 years

The parameters to be considered for purposeful sampling include: (1) clinical area, (2) impact of the paper (journal impact and citation analysis), (3) type of intervention and comparator (eg, a novel invasive surgical procedure), (4) trial design (eg, pragmatic or explanatory) and (5) country of trial conduct (single country or international). The index trials will be identified from a shortlist of 20 top cited major surgical RCTs published between 2006 and 2016 in English. This time frame will allow trials to accrue citations, while also focusing on evidence still relevant to current practice. RCTs will be identified in the Web of Science and Scopus search engines. These tools combine primary search functionality with citation tracking to count and identify articles citing selected RCTs. In the first instance, Scopus will be searched using the following terms within the title field: (randomi*ed trial) AND (surg* or operat* or laparoscop* or *ectomy or *otomy or *plasty or *rrhaphy or *ostomy or *pexy or fusion or

arthrodesis or arthroscop* or ((internal or external) and fixation) or (caesarean section) or (bypass and (artery or graft)) or ((repair or replace*) AND (hernia or aneurysm or valve or fistula or hip or knee or ankle or shoulder or elbow)) or ((resection or excision or dissection) AND (axillary or anterior or rect* or liver* or hepatic or gastric or colon* or colorectal* or (lymph node*) or (small bowel)))) AND NOT (chemotherap*). Search results will be sorted in descending numerical order of total citation count, as only a subset of citing articles will be included for analysis. Article titles and abstracts will be screened to identify the 20 most highly cited surgical RCTs (in descending order of number of citations from the Scopus database), involving 2 or more groups, with comparison between at least one surgical technique and another invasive treatment. The search will be repeated using the Web of Science search engine, and a consolidated list created. The shortlist will be reviewed by all authors to decide which RCTs will be examined. The full text of the selected trial reports will be retrieved. Based on pilot work, future larger scale application of this method may permit sampling of many RCTs across the entire range of criteria described above.

### Identify and systematically sample the articles citing the selected RCTs

Articles citing each trial will be identified in two ways: using Web of Science and Scopus citation tracking; and using the Altmetric.com 'bookmarklet' to identify social media and web-based articles.[18]

The Web of Science and Scopus search engines provide a list of all publications citing an individual article. There are 41 different types of document indexed by Web of Science, and 15 types indexed by Scopus (see online supplementary material table 1). From the list of articles citing the selected RCTs, the following types will be included from the Web of Science: discussion, editorial material and letter. The following documents will be selected from Scopus: editorial and letter. All other article types will be excluded. These types have been chosen because the authors consider that the selected document types are likely to contain a higher proportion of author opinions than other data sources. If initial searches yield insufficient data, this aspect of the method will be reviewed by the project team to widen the search strategy and include a larger sample of relevant articles. All citing articles of the selected types will be analysed. This will include articles that respond directly to the index RCTs, for example, commenting on its design or findings, as well as those that cite the trials as part of broader or apparently unrelated discussions. This approach will help investigate the understanding and utilisation of evidence from RCTs in the broad setting of evidence-based medicine.

Further online articles citing the included RCTs will be identified using Altmetric.com. Altmetric.com is one available source of alternative metrics, or 'altmetrics'. Altmetrics are an alternative to conventional citation-based metrics that specifically examine online attention given to academic material.[19] They include citations on Wikipedia and in policy documents, as well as discussions on blogs, mainstream media and social networks such as Twitter. Combination with traditional citations in the indexed scientific literature will help create a broad picture of the articles citing surgical RCTs.

### Undertake in-depth qualitative analysis and identify emerging themes

Eligible articles in both the traditional academic press and the online environment will be compiled in electronic format. Given the focused objectives of this project, a formal grounded theory approach will not be adopted.[20] However, thematic analysis will be performed using the constant comparison technique, adopted from grounded theory, to identify stated and latent themes and create a rich description of the data.[21] All analysis will be conducted using NVivo for Windows V.11.4 (QSR International, Australia).

An initial sample of five articles will be carefully read and reread in detail by BEB—a postdoctoral surgeon-researcher who has previous experience of qualitative research. Individual sections of text will be coded to summarise content and meaning. As the project has very specific objectives, some of the likely codes/themes can be anticipated, but no a priori coding criteria will be specified, and all content of articles will be coded (although in more/less depth, in accordance with the project objectives). This will ensure that findings will be grounded in the data as much as possible. Coded text will be grouped into themes, creating hierarchies of superordinate and subordinate nodes. The initial five articles will be independently double coded by LR, a non-clinical lecturer in qualitative health sciences. BEB and LR will meet to discuss and review the initial five coded articles and seek to resolve any discrepancies in coding. Double coding will be performed on a further five articles if there are 'significant' discrepancies (which will be judged at the time of analysis but is likely to constitute discrepancies that extend beyond differences in how codes are named). The second round of double coding, if conducted, will be followed by another review between coders. Thereafter, provided there is satisfactory agreement, further analysis will be conducted by BEB. An iterative approach will continue, with further analysis until thematic saturation is achieved, with no new themes identified in five articles (or until the project ends). An evolving descriptive account will be created, circulated and discussed at regular face-to-face meetings to revise and refine themes as they emerge. BEB will also make reflexive notes on the coding process. The study team includes a senior trials methodologist and surgeon (JMB) and a senior medical anthropologist (HL). The extensive experience in qualitative research and surgical trials of this multidisciplinary team will help create a coherent thematic structure that is meaningful to all stakeholders. In the final analysis, emergent themes will be organised and categorised to best explain the observed findings.

### Summarise validity and reporting of included RCTs

After completing the qualitative analysis, the selected RCTs will be appraised using established, validated tools designed to assess their reporting, internal and external validity. This will involve collection of data using the following tools: Consolidated Standards of Reporting Trials for Non-Pharmacological Treatments checklist[22]; Cochrane Risk of Bias Tool[7]; Pragmatic-Explanatory Continuum Indicator Summary 2 tool[23]; Context and Implementation of Complex Interventions checklist.[24] Where necessary, protocol papers or documents will be sought to collect a complete dataset if possible. Simple study characteristics and publication information will also be compiled.

### Combine results to develop deeper understanding of how trials are understood and the relationship with trial quality

The results of the thematic analysis described in step 3 above will be compared with the included RCTs' methodological and reporting quality, as determined in step 4 described above. The results of both analyses will be collated to explore perceptions and understanding of authors that use and cite RCT evidence in relation to the methodological and reporting quality of the relevant index RCT.

### Develop and refine the methodology

Pilot work will be conducted to establish the feasibility of using this new method. It is anticipated that the experience gained will result in iterative improvements to the method before future, larger scale application to investigate published reactions to other RCTs in surgery or in other areas of healthcare. For example, a framework may be produced for use in further reviews to categorise data emerging from the qualitative analysis. Future developments are likely to include purposive sampling of multiple RCTs using a range of different criteria to identify common themes and differences across different medical fields. It may also be useful to include a greater range of citing articles, such as other trials and review articles, rather than restricting the qualitative analysis to editorial, discussion and letter article types.

### Create recommendations to inform future design, conduct, reporting and implementation of surgical RCTs

The findings of this study will lead to recommendations for future trial design and reporting to anticipate and avoid common criticisms and misunderstandings. Beyond RCT design, this study may also yield insights to optimise translation of evidence from high-quality RCTs into day-to-day clinical practice by helping identify common perceptual barriers to the uptake of research evidence. For example, study findings may indicate a need for focused education for surgeons to improve understanding of when and how to implement new interventions based on RCT findings. This may improve efficiency in the generation and application of evidence on the clinical utility of surgical interventions, to the benefit of patients, clinicians and the healthcare system.

### Patient and public involvement

Patients and members of the public were not involved in any aspect of the design of this study.

## ETHICS AND DISSEMINATION

This protocol paper describes a new means of understanding how clinicians and academics respond to, and use, published trial data. The findings will be disseminated through conference presentation and peer-reviewed publication, as well as social media channels. If successful, the results of pilot work will be used to inform a future, larger scale application of the method which will also be published and presented, and promoted through social media.

This method has the potential to generate new knowledge both to improve RCT design and reporting, and to optimise translation of knowledge from RCTs into everyday clinical practice. In the future, it may be complemented by other research techniques such as in-depth interviews, to build on the findings and explore the issues identified. Any developments in understanding will be used by the trial team and our network to help improve design and implementation of trials that will be hosted or supported by the team and the associated Bristol Clinical Trials and Evaluation Unit.

**Acknowledgements** We would like to thank Cath Borwick, Information Specialist at the University of Bristol, for helping develop the search strategy and highlighting other resources for identifying responses to trials.

**Contributors** BEB and JMB conceived the study. BEB drafted the protocol. BEB, LR, HL and JMB provided intellectual input into improving the study design and revising the protocol. All authors have contributed to and approved the final manuscript.

**Funding** BEB is supported by the National Institute for Health Research (NIHR). JMB is an NIHR senior investigator. This work was supported by the NIHR Biomedical Research Centre at the University Hospitals Bristol NHS Foundation Trust and the University of Bristol. This work was undertaken with the support of the MRC ConDuCT-II (Collaboration and innovation for Difficult and Complex randomised controlled Trials In invasive procedures) Hub for Trials Methodology Research (MR/K025643/1) and support from the Royal College of Surgeons of England Bristol Surgical Trials Centre.

**Disclaimer** The views expressed are those of the authors and not necessarily those of the NHS, the NIHR or the Department of Health. The funders had no role in developing the protocol.

**Competing interests** None declared.

**Patient consent** Not required.

**Ethics approval** This study involves secondary use of written material that is already publicly available and does not require ethical review.

**Provenance and peer review** Not commissioned; externally peer reviewed.

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
