## [Reviewer comments · BMJ Open]

ARTICLE DETAILS

TITLE (PROVISIONAL)	Evidence into practice: protocol for a new mixed methods approach to explore the relationship between trials evidence and clinical practice through systematic identification and analysis of articles citing randomised controlled trials.
AUTHORS	Byrne, Benjamin; Rooshenas, Leila; Lambert, Helen; Blazeby, Jane

VERSION 1 – REVIEW

REVIEWER	Jodi Schneider School of Information Sciences, University of Illinois at Urbana-Champaign, USA
REVIEW RETURNED	05-May-2018

GENERAL COMMENTS	This protocol aims to understand the concerns expressed about the quality or relevance of RCT evidence to surgical practice by analyzing editorials, discussions, letters, social media, and web-based articles that reference a sample of 20, purposefully selected RCTs published from 2006-2016. Authors will use qualitative coding to identify "factors and issues of concern to clinicians when interpreting the results of an RCT and relating them to their own practice" which could help "develop recommendations to inform future design, conduct, reporting and implementation of surgical RCTs" that would help move evidence into practice. To my knowledge, this is a novel approach, which should provide valuable results. Rationale for using this method first in surgery is sensible due to the "evidence that surgeons tend to value clinical experience and individualism over research findings(18) and tend to resist changing established patterns of practice(19)." I commend the protocol for describing the team. I was reassured to see that the main coder and first author has previously used qualitative coding to identify 3 themes from an interview study reported in Chapter 10 of his PhD thesis http://hdl.handle.net/10044/1/34688, and conducted with the same tools (NVivo 10 for Windows). The second coder also has a background in qualitative health sciences, and the extended team brings complementary expertise in the content area (surgery), trials methodology, and medical anthropology. When you document this methodology for others to use for their work, I encourage you to emphasize the relevant competencies. It's unclear whether your project is underway or still in the planning processes (you mention "past 10 years" for literature in one place, and 2006 to 2016 in another; and citing literature will appear with various delays). Inclusion of an information professional or information scientist could be helpful for filtering search materials,
---

	drawing on citation analysis theory, etc., especially if you are still selecting the literature to be used. If your health sciences library does not have librarians, information scientists, etc. with an appropriate background, you might get Andrew Booth, Reader in Evidence Based Information Practice and Director of Information at the University of Sheffield <https://www.sheffield.ac.uk/scharr/sections/ir/staff/booth_a>, who has expertise in qualitative synthesis, evidence-based practice, and related areas. Another area where an information specialist could help is in making sense of emerging trends in citation practice, such as the altmetrics services available. Note that "altmetrics" is not a brand name (see e.g. http://altmetrics.org/manifesto/) and I would describe the bookmarklet using "Altmetric.com" for clarity. Altmetric.com is a reasonable provider, but note that they only provide data from 2011 forwards; they rely on DOIs (so coverage may depend on the journal and year); and you may want to request (free) access to their academic research data service. As the Altmetrics 2018 conference will be held nearby in London, you may wish to attend to connect with that community: http://altmetrics.org/altmetrics18/ In any case, within the manuscript please use references or footnotes to indicate that you are talking about Altmetric.com. Google Scholar data is harder to harvest than data from the other sources you are using but can be a good place to find grey literature and emerging sources. The current abstract is not clear enough. Even in the introduction, I think you could mention commentaries (editorials, discussions, letters); the mention of "citing" literature obscures what you are doing. You do not mention the "constant comparison approach" anywhere else in the body of the protocol; either remove it or add a discussion, with a citation. If you already can identify, exemplify, or describe "generic issues" or "features specific to surgery", you could mention that. In Design (page 7), you mention "journal bibliography" impact -- it's not clear what you mean by this. On page 8, you may want to explain the time period (e.g. last 10 years? or having time for citation elapse?). Explain why sorting by total citation count makes sense, when you are going to filter out many of these citations. On page 10, I wanted to know the tools for thematic analysis right away; you do cover this, but consider moving that information earlier in the protocol. A key issue within your process is the validity checking (page 11); how you choose the initial 5 articles may make a difference. You indicate that you'll continue double-coding beyond 5 articles; I would encourage you to solicit feedback (if not double-coding) from the second coder (due to qualitative expertise) whenever new codes emerge. While I think you would do that anyway (given your qualitative expertise), it's particularly important to better document this type of process and decision-making if you plan to disseminate your methods since not all health and medical researchers have access to qualitative experts; this could be undertaken in the "develop and refine the methodology" process you mention. The appraisal (and thanks for the list of tools that you will use) will be an important complementary approach.
--	--

	Under ethical considerations, please replace "available in the public domain" with "publicly available". This is an important distinction; an information specialist, again, could help here. In the funding, you might provide (NIHR) as an acronym in the first sentence since it's not universally understood, but is used in the second sentence. A finer point in the bibliography: use DOIs consistently when available (rather than URLs that contain the DOI in e.g. 20, 22, 23 among others). In these cases, citation date should not be needed.
--	---

REVIEWER	Eivind Engebretsen University of Oslo, Norway
REVIEW RETURNED	24-Jun-2018

GENERAL COMMENTS	This protocol presents an innovative approach examining how researchers understand, contextualise and use evidence from RCTs, through analysis of citing articles. The method is original and thoroughly described. However, the theoretical basis for the qualitative analysis is weak. The coding process should be described in more detail with reference to current literature within grounded theory. Overall, this is a timely, innovative and well-structured protocol that deserves to be published.
--

VERSION 1 – AUTHOR RESPONSE

Reviewer 1:

This protocol aims to understand the concerns expressed about the quality or relevance of RCT evidence to surgical practice by analyzing editorials, discussions, letters, social media, and web-based articles that reference a sample of 20, purposefully selected RCTs published from 2006-2016. Authors will use qualitative coding to identify "factors and issues of concern to clinicians when interpreting the results of an RCT and relating them to their own practice" which could help "develop recommendations to inform future design, conduct, reporting and implementation of surgical RCTs" that would help move evidence into practice. To my knowledge, this is a novel approach, which should provide valuable results. Rationale for using this method first in surgery is sensible due to the "evidence that surgeons tend to value clinical experience and individualism over research findings(18) and tend to resist changing established patterns of practice(19)."

I commend the protocol for describing the team. I was reassured to see that the main coder and first author has previously used qualitative coding to identify 3 themes from an interview study reported in Chapter 10 of his PhD thesis <<http://hdl.handle.net/10044/1/34688>>, and conducted with the same tools (NVivo 10 for Windows). The second coder also has a background in qualitative health sciences, and the extended team brings complementary expertise in the content area (surgery), trials methodology, and medical anthropology. When you document this methodology for others to use for their work, I encourage you to emphasize the relevant competencies.

Thank you for this helpful comment. We have added a sentence into the methods on page 10 to emphasise the experience and multidisciplinary nature of the study team.

It's unclear whether your project is underway or still in the planning processes (you mention "past 10 years" for literature in one place, and 2006 to 2016 in another; and citing literature will appear with various delays). Inclusion of an information professional or information scientist could be helpful for filtering search materials, drawing on citation analysis theory, etc., especially if you are still selecting the literature to be used. If your health sciences library does not have librarians, information scientists, etc. with an appropriate background, you might get Andrew Booth, Reader in Evidence Based Information Practice and Director of Information at the University of Sheffield https://www.sheffield.ac.uk/scharr/sections/ir/staff/booth_a, who has expertise in qualitative synthesis, evidence-based practice, and related areas.

We apologise for the omission from our previous submission – we discussed the study with an information scientist at the University of Bristol, who helped develop the literature search strategy, pointed us to use Scopus as well as Web of Science, and suggested including altmetrics. We have now added an acknowledgement section to the end of the manuscript to recognise this input.

Another area where an information specialist could help is in making sense of emerging trends in citation practice, such as the altmetrics services available. Note that "altmetrics" is not a brand name (see e.g. <http://altmetrics.org/manifesto/>) and I would describe the bookmarklet using "Altmetric.com" for clarity. Altmetric.com is a reasonable provider, but note that they only provide data from 2011 forwards; they rely on DOIs (so coverage may depend on the journal and year); and you may want to request (free) access to their academic research data service. As the Altmetrics 2018 conference will be held nearby in London, you may wish to attend to connect with that community: <http://altmetrics.org/altmetrics18/> In any case, within the manuscript please use references or footnotes to indicate that you are talking about Altmetric.com. Google Scholar data is harder to harvest than data from the other sources you are using but can be a good place to find grey literature and emerging sources.

Thank you again for this useful advice. As above, we have been designing this study with input from a local information scientist, who directed us to use altmetrics. We have revised the manuscript to make it clearer that we will use the specific, branded Altmetric.com bookmarklet. We are cognisant of the relative recency of Altmetric.com and that this is in tension with using high citation count to select individual studies for inclusion – as accruing traditional citations occurs over a period of years. However, we wanted to include this in the study design as this can only become more important in the future.

The current abstract is not clear enough. Even in the introduction, I think you could mention commentaries (editorials, discussions, letters); the mention of "citing" literature obscures what you are doing. You do not mention the "constant comparison approach" anywhere else in the body of the protocol; either remove it or add a discussion, with a citation. If you already can identify, exemplify, or describe "generic issues" or "features specific to surgery", you could mention that.

Thank you. We have revised the abstract and we believe that these changes make the project and its details clearer to the reader.

In Design (page 7), you mention "journal bibliography" impact -- it's not clear what you mean by this. On page 8, you may want to explain the time period (e.g. last 10 years? or having time for citation elapse?). Explain why sorting by total citation count makes sense, when you are going to filter out many of these citations. On page 10, I wanted to know the tools for thematic analysis right away; you do cover this, but consider moving that information earlier in the protocol.

Journal bibliography has been changed to journal impact. As you suggest, we have described the rationale for the timeframe chosen, and our strategy to use the highest cited articles because of the subsequent sampling of a subset of the total citing articles. We have moved reference to nVivo to the first paragraph describing the thematic analysis.

A key issue within your process is the validity checking (page 11); how you choose the initial 5 articles may make a difference. You indicate that you'll continue double-coding beyond 5 articles; I would encourage you to solicit feedback (if not double-coding) from the second coder (due to qualitative expertise) whenever new codes emerge. While I think you would do that anyway (given your qualitative expertise), it's particularly important to better document this type of process and decision-making if you plan to disseminate your methods since not all health and medical researchers have access to qualitative experts; this could be undertaken in the "develop and refine the methodology" process you mention.

We have revised the relevant methods section to clarify the iterative and collaborative process involved in the analysis, with discussion, refinement and revision of themes, using the experience of the multidisciplinary study team, as they emerge.

The appraisal (and thanks for the list of tools that you will use) will be an important complementary approach.

Under ethical considerations, please replace "available in the public domain" with "publicly available". This is an important distinction; an information specialist, again, could help here.

We have revised this as suggested.

In the funding, you might provide (NIHR) as an acronym in the first sentence since it's not universally understood, but is used in the second sentence.

Also revised as suggested, thank you.

A finer point in the bibliography: use DOIs consistently when available (rather than URLs that contain the DOI in e.g. 20, 22, 23 among others). In these cases, citation date should not be needed.

We have changed the citation style file to include DOIs and follow the required formatting for BMJ Open.

Reviewer 2:

This protocol presents an innovative approach examining how researchers understand, contextualise and use evidence from RCTs, through analysis of citing articles. The method is original and thoroughly described. However, the theoretical basis for the qualitative analysis is weak. The coding process should be described in more detail with reference to current literature within grounded theory. Overall, this is a timely, innovative and well-structured protocol that deserves to be published.

Thank you for these encouraging comments. In response to concerns about the theoretical basis for the analysis, we have amended the relevant section of the methods on pages 9-10. We have clarified that while we have adopted techniques from grounded theory, we are not undertaking a formal grounded theory approach due to a number of reasons, including the focused pre-specified aims/objectives of the study and the fact that the principal coder is a surgeon who will inevitably bring some preconceptions to the analysis. Given this, we have also added a sentence highlighting that BEB will make reflexive notes during the analysis to enhance the transparency of the analysis. We have also provided some more detail regarding the process of developing themes within the multidisciplinary study team.

We hope that the changes satisfactorily address the reviewer's concerns and we are encouraged by the recommendation for publication.

VERSION 2 – REVIEW

REVIEWER	Jodi Schneider University of Illinois at Urbana-Champaign, USA
REVIEW RETURNED	08-Aug-2018

GENERAL COMMENTS	In the abstract, I suggest you remove "all" and replace "systematically identify all citing articles" with "systematically identify citing articles" . This is because no source is completely comprehensive; in fact Web of Science and Scopus both have purposeful, selective coverage. So in fact, you will "systematically identify all citing articles covered in Web of Science or Scopus". Thank you for mentioning the Altmetric.com bookmarklet more clearly on page 8; I would suggest that you replace "Altmetric" with "Altmetric.com" as well on page 9. The general category is
---

	"altmetrics" but you want to talk PARTICULARLY about the Altmetric.com implementation. You may wish to cite Priem J, Taraborelli D, Groth P, Neylon C. Altmetrics: A manifesto. http://altmetrics.org/manifesto/ in that paragraph; but in any case, please clearly distinguish when you are using the particular (Altmetric.com) vs. the general (altmetrics), by avoiding using the singular "Altmetric" as a stand-alone noun. The additions on page 10 regarding thematic saturation, evolution, and the study team, are useful; thank you for those. The discussion of the potential for the method to generate new knowledge, in "ethics and dissemination" is also a very useful addition. Thank you for adding your information specialist to the acknowledgements. In a systematic review, developing the search strategy would count for authorship though that need not be appropriate in this case. REFERENCES The DOIs can be added to reference #9: 10.1016/S0140-6736(04)17670-8 I believe that the journal style (3 authors + et al.) hides relevant information (especially last author) but their inappropriate choice is beyond your control. As links were last checked about 6 months ago at original submission, you may wish to check them and update the date when you make the final submission. I very much look forward to the results of your protocol!
--	---

REVIEWER	Eivind Engebretsen University of Oslo, Norway
REVIEW RETURNED	20-Aug-2018

GENERAL COMMENTS	Thank you for the opportunity to review your revised paper. I believe you have addressed much of the previous reviewers comments and suggestions. In my view, the paper is now ready for publishing.
--

VERSION 2 – AUTHOR RESPONSE

Reviewer(s)' Comments to Author:

Reviewer: 1

Reviewer Name: Jodi Schneider

Institution and Country: University of Illinois at Urbana-Champaign, USA

Please state any competing interests or state 'None declared': None declared

Please leave your comments for the authors below

In the abstract, I suggest you remove "all" and replace "systematically identify all citing articles" with "systematically identify citing articles" . This is because no source is completely comprehensive; in

fact Web of Science and Scopus both have purposeful, selective coverage. So in fact, you will "systematically identify all citing articles covered in Web of Science or Scopus".

This has been changed as suggested.

Thank you for mentioning the Altmetric.com bookmarklet more clearly on page 8; I would suggest that you replace "Altmetric" with "Altmetric.com" as well on page 9. The general category is "altmetrics" but you want to talk PARTICULARLY about the Altmetric.com implementation. You may wish to cite Priem J, Taraborelli D, Groth P, Neylon C. Altmetrics: A manifesto. <http://altmetrics.org/manifesto/> in that paragraph; but in any case, please clearly distinguish when you are using the particular (Altmetric.com) vs. the general (altmetrics), by avoiding using the singular "Altmetric" as a stand-alone noun.

This paragraph on page 9 has been amended to clearly distinguish between altmetrics in general, and Altmetric.com. We now also cite the useful article suggested.

The additions on page 10 regarding thematic saturation, evolution, and the study team, are useful; thank you for those.

The discussion of the potential for the method to generate new knowledge, in "ethics and dissemination" is also a very useful addition.

Thank you for adding your information specialist to the acknowledgements. In a systematic review, developing the search strategy would count for authorship though that need not be appropriate in this case.

REFERENCES

The DOIs can be added to reference #9: 10.1016/S0140-6736(04)17670-8

Added, thanks.

I believe that the journal style (3 authors + et al.) hides relevant information (especially last author) but their inappropriate choice is beyond your control.

As links were last checked about 6 months ago at original submission, you may wish to check them and update the date when you make the final submission.

Weblinks checked and today's date applied.

I very much look forward to the results of your protocol!

Thank you for taking such time to review our manuscript so thoroughly.

Reviewer: 2

Reviewer Name: Eivind Engebretsen

Institution and Country: University of Oslo, Norway

Please state any competing interests or state 'None declared': None declared

Please leave your comments for the authors below

Thank you for the opportunity to review your revised paper. I believe you have addressed much of the previous reviewers comments and suggestions. In my view, the paper is now ready for publishing.

Thank you.